# Morphological and Molecular Identification of *Ulva* spp. (Ulvophyceae; Chlorophyta) from Algarrobo Bay, Chile: Understanding the Composition of Green Tides

**DOI:** 10.3390/plants13091258

**Published:** 2024-04-30

**Authors:** Javiera Mutizabal-Aros, María Eliana Ramírez, Pilar A. Haye, Andrés Meynard, Benjamín Pinilla-Rojas, Alejandra Núñez, Nicolás Latorre-Padilla, Francesca V. Search, Fabian J. Tapia, Gonzalo S. Saldías, Sergio A. Navarrete, Loretto Contreras-Porcia

**Affiliations:** 1Departamento de Ecología y Biodiversidad, Facultad de Ciencias de la Vida, Universidad Andres Bello, Santiago 8370251, Chile; jaavimuti@gmail.com (J.M.-A.); mramirezcasali@gmail.com (M.E.R.); meynardster@gmail.com (A.M.); benjamin.pinilla.rojas11@gmail.com (B.P.-R.); gnunez.alejandra@gmail.com (A.N.); 2Centro de Investigación Marina Quintay (CIMARQ), Facultad de Ciencias de la Vida, Universidad Andres Bello, Quintay 2531015, Chile; 3Instituto Milenio en Socio-Ecología Costera (SECOS), Santiago 8370251, Chile; phaye@ucn.cl (P.A.H.); nlatorrepadilla@gmail.com (N.L.-P.); fran.search.t@gmail.com (F.V.S.); gsaldias@ubiobio.cl (G.S.S.); snavarrete@bio.puc.cl (S.A.N.); 4Center of Applied Ecology and Sustainability (CAPES), Santiago 8331150, Chile; 5Departamento de Biología Marina, Facultad de Ciencias del Mar, Universidad Católica del Norte, Coquimbo 1780000, Chile; 6Estación Costera de Investigaciones Marinas, Las Cruces, Facultad de Ciencias Biológicas, Pontificia Universidad Católica de Chile, Santiago 8331150, Chile; 7Millenium Nucleus for Ecology and Conservation of Temperate Mesophotic Reef Ecosystems (NUTME), Las Cruces, Pontificia Universidad Católica de Chile, Santiago 8331150, Chile; 8Departamento de Oceanografía, Facultad de Ciencias Naturales y Oceanográficas, Universidad de Concepción, Concepción 4070386, Chile; ftapia@oceanografia.udec.cl; 9Centro de Investigación Oceanográfica COPAS Coastal, Universidad de Concepción, Concepción 4070386, Chile; 10Departamento de Física, Facultad de Ciencias, Universidad del Bío-Bío, Concepción 4051381, Chile

**Keywords:** green tides, *ITS1*, phylogeny, taxonomy, *tuf*A, *Ulva stenophylloides*, *Ulva uncialis*, *Ulva compressa*, *Ulva aragoensis*, *Ulva australis*

## Abstract

Green algae blooms of the genus *Ulva* are occurring globally and are primarily attributed to anthropogenic factors. At Los Tubos beach in Algarrobo Bay along the central Chilean coast, there have been blooms of these algae that persist almost year-round over the past 20 years, leading to environmental, economic, and social issues that affect the local government and communities. The objective of this study was to characterize the species that form these green tides based on a combination of ecological, morpho-anatomical, and molecular information. For this purpose, seasonal surveys of beached algal fronds were conducted between 2021 and 2022. Subsequently, the sampled algae were analyzed morphologically and phylogenetically using the molecular markers *ITS1* and *tuf*A, allowing for the identification of at least five taxa. Of these five taxa, three (*U. stenophylloides*, *U. uncialis*, *U. australis*) have laminar, foliose, and distromatic morphology, while the other two (*U. compressa*, *U. aragoensis*) have tubular, filamentous, and monostromatic fronds. Intertidal surveys showed that *U. stenophylloides* showed the highest relative coverage throughout the seasons and all intertidal levels, followed by *U. uncialis*. Therefore, we can establish that the green tides on the coast of Algarrobo in Chile are multispecific, with differences in relative abundance during different seasons and across the intertidal zone, opening opportunities for diverse future studies, ranging from ecology to algal biotechnology.

## 1. Introduction

Over the past few decades, the proliferation and subsequent beaching of unattached green macroalgae have become a more common and widespread occurrence worldwide [1,2], suggesting that such events may be related to anthropogenically altered ocean conditions and hence another component of global change [3,4,5,6]. These so-called green tides typically include several species of green macroalgae of the genera *Cladophora*, *Spirogyra*, and, especially, *Ulva*, which grow and detach from the hard substratum, accumulating high biomass while suspended in the water column, causing large and detrimental ecological and socioeconomic effects on coastal ecosystems [1,2,3,4,5,6,7,8,9,10]. The largest and probably best-studied green tide has occurred every summer in China since 2007, which at its peak caused the accumulation of over one million tons of green algal fronds on the beaches around the city of Qingdao, inflicting costs of over USD 200 million on the local government for their removal [11,12]. These Yellow Sea green tides have been persistently dominated by one species: *Ulva prolifera* [13]. On the shores of Brittany, France, green algal tides dominated by *Ulva* have occurred since the 1970s. The detrimental effects they cause on coastal ecosystems, along with the danger to human health by the emanation of noxious gas from decomposing algae, led to the creation of an Anti-Algal plan by the French government in 2010 [6].

The proximate causes of green tides are still not well understood, although they are probably multi-causal and likely vary in importance among regions. Commonly, they involve coastal eutrophication generated by increased loads of nitrogen and phosphorous derived from agriculture and cities [6,9]. Coastal eutrophication and marine contamination from heavy metals have significantly increased worldwide over the past few decades (e.g., [14,15]), causing adverse effects on marine ecosystems. Both disturbances are linked to human activities, including municipal contamination, agricultural activities, and oil discharges into the coastal ocean [14,15,16,17,18,19]. Other factors that have been connected to green tides include alteration of habitat availability by aquaculture and other anthropic activities [11]. Among the species commonly considered pollution bioindicators, *Ulva* species have been highlighted for rapidly increasing their abundance under certain forms of pollution (e.g., heavy metals) and for being able to rapidly uptake nutrients even when available in pulses [20]. Indeed, in several species of this genus, diverse physiological mechanisms have been described that enable them to inhabit polluted marine environments (e.g., [21,22,23]).

In Algarrobo Bay, located on the central coast of Chile, green tides dominated by species of *Ulva* have occurred yearly since 2002 along ca. 2 km of shoreline at the southwest end of the bay (Municipal Report, Figure 1a). The beached algae have significantly deteriorated beach conditions, reducing tourism, decreasing property value around the area, and forcing the municipality to remove the algae, preventing hazards to human health. Although the extent of Algarrobo’s green tides is considerably smaller than those observed in other regions of the world, they are an important source of socioecological conflict and a cause of ecosystem deterioration, both directly and indirectly. Since green tides are expected to increase their frequency in years to come, potentially affecting larger sections of the coast, it is crucial to know which species are involved and to improve our understanding of the proximate causes of these blooms to inform monitoring and management strategies. Therefore, the first step towards this goal is to determine which species of *Ulva* compose the Algarrobo Bay green tides and their relative coverage throughout the seasons and intertidal levels.

In Chile, the classical taxonomy of this genus (principally at the morphological level) was revised in the 1990s in both its foliose (*Ulva* spp.) and tubular (*Enteromorpha* spp.) forms [24,25,26]. Records based on classical morphological descriptions and phylogenetic inferences are scant, with only one published study that describes *Ulva australis* along the shores of Iquique (20°16′–13′ S; 70°7′–9′ W). This species is cosmopolitan [27] and commonly found in green tides worldwide, showing a seasonal pattern in which the attached biomass increases from winter to spring but declines at higher temperatures during summer and autumn [28]. However, there is no robust information on this species forming part of the green tides registered in Algarrobo Bay or any other section of the Chilean coast.

In addition to classical taxonomy, molecular phylogenetic analyses have proven to be a useful aid in species identification through a barcoding approach [29,30]. This is particularly applicable to *Ulva* species, which have a simple morphology and can display extensive plasticity in morphological traits (e.g., [31,32,33]), leading to cryptic species that are difficult to detect using morphology-based identification alone. By using phylogenetic analyses, DNA markers have facilitated species determination within *Ulva* (e.g., [34,35,36]). For the genus *Ulva*, sequences of the elongation factor Tu (*tuf*A) of the chloroplast genome have been shown to exhibit adequate variation for species resolution and are widely used to assess species diversity (e.g., [34,35,36,37,38,39,40,41,42,43,44,45,46]). Other markers that have proven useful for species identification and often complement data from *tuf*A or other chloroplast sequences are the Cytochrome Oxidase I (*COI*) gene encoded in the mitochondrial genome (e.g., [47]) and sequences of internal transcribed spacer *ITS* from the nuclear genome [32,34,35,36,37,38,42].

Here, we assess the diversity of *Ulva* species from Algarrobo Bay using both a classical taxonomic analysis and phylogenetic species determination using two DNA markers. We used DNA sequences from genomes of each cellular compartment: nuclear *ITS1* and chloroplast *tuf*A. We performed phylogenetic and species determination analyses based on 43 GenBank sequences and 55 newly obtained sequences of the specimens identified by morphological characterization. Furthermore, we determined the distribution along the intertidal zone of different ‘morphological species’ and their abundance over a yearly cycle. The results of this study will provide new insights into *Ulva* diversity associated with the green tides in Chile and the South Pacific.

## 2. Results

Analyses of the data recorded for both the external and internal morphological characteristics of the morphotypes recognized in the field, in combination with the molecular data obtained via the analysis of the molecular markers *ITS1* and *tuf*A, have allowed us to corroborate the presence of five taxa of the genus *Ulva* (Appendix A). Of these five taxa, three are of laminar, foliose, and distromatic morphology, corresponding to the species *Ulva stenophylloides* L.G. Kraft, Kraft & R.F. Waller (Figure 1b), *Ulva uncialis* (Kützing) Montagne (Figure 1c), and *Ulva australis* Areschoug. The other two are tubular, filamentous, and monostromatic fronds corresponding to the species *Ulva compressa* Linnaeus (Figure 1d) and *Ulva aragoensis* (Bliding) Maggs (Figure 1e) (Table 1). Only one of the specimens sampled was genetically corroborated as *Ulva australis*, which precluded an exhaustive morphological analysis for this species. The external morphological features of this individual (Appendix A) are consistent with those described by [39] for *U. australis* from northern Chile (Iquique).

### 2.1. Ulva *spp.* Covers

The data gathering revealed that *U. stenophylloides* consistently dominates with the highest relative cover nearly all seasons and across intertidal levels, averaging 53%, at Los Tubos beach. During winter, its cover was relatively similar to that of *U. uncialis* across the intertidal zones, while other species of the *Ulva* genus were scarcely present. In spring, *U. stenophylloides* exhibited the highest cover in both the high and mid-intertidal zones, followed by other *Ulva* species. Across both summer and spring, *U. stenophylloides* maintained the highest cover across all intertidal levels. *U. uncialis* showed similar cover across intertidal zones throughout seasons, with the highest cover observed in winter (Figure 2).

### 2.2. Description of Specimens

#### 2.2.1. *Ulva stenophylloides* L.G. Kraft, Kraft & R.F. Waller

The thallus was deep green to light green in color, variable in shape from orbicular in outline to broadly foliose, and sometimes laciniate (Figure 3). It is attached to the substrate by a small, round disk 3.01 ± 0.64 mm in diameter, which extends into a small, solid, almost inconspicuous stipe. The fronds are smooth to the touch, with few perforations, smooth to slightly wavy margins, and largely serrated, although a smoother edge is observed in the basal part (Figure 3).

The thallus, on average, measures a total length of 8.04 ± 6.37 cm with a basal width of 0.7 ± 0.3 cm. In the middle zone, the width of the thallus is 2.01 ± 1.1 cm, and at the top, it measures 1.86 ± 1 cm. The fronds are distromatic, with an average midzone thickness of 36.74 ± 7.67 µm. The cells are arranged in a disorderly manner in a superficial view (Figure 4a). In a cross-section, they are elongated rectangles, measuring on average 11.53 ± 3.6 µm high × 7.35 ± 1.7 µm in diameter (Figure 4b). The chloroplasts occupy a large part of the cell, are cup-shaped, and present between one and two pyrenoids per cell. There is an interlayer space with visible mucilage and an average width in the middle zone of 6.6 ± 1 µm.

This species has only been previously recorded from Australia, New Zealand, and South Africa [27,48]; therefore, this is the first record for Chile.

#### 2.2.2. *Ulva uncialis* (Kützing) Montagne

The thallus is of variable morphology; some thalli have an elongated laminar shape, tending to be lanceolate, and others have short and folded fronds (Figure 5). The fronds of *Ulva uncialis* are dark green in color and adhere to the substrate through a conspicuous, circular-shaped adhesive disk 3 ± 1.53 mm in diameter. The fronds are smooth-textured, with perforations and a conspicuous thickening at the basal part that decreases towards the top. The frond edges are smooth to wavy, sometimes with denticulations. The basal part of the thallus has short excrescences.

On average the thalli measure 11.5 ± 4.83 cm in total length, 0.68 ± 0.56 cm in basal width, 5.3 ± 2.3 cm wide in the middle portion of the thallus, and 3.8 ± 2.2 cm wide at the upper end of the thallus. The fronds are distromatic, with an average thickness of 109 ± 59 µm in the cross-section of the midzone (Figure 6a,b). The cells vary in shape from irregularly round to pyriform, reaching an average height of 24 ± 8.2 µm and an average diameter of 9.02 ± 2.1 µm in the middle zone of the thallus. The interlayer space measures 10.1 ± 5.4 µm (Figure 6a).

The shape of the chloroplasts is variable from oval to round. It presents between one and two pyrenoids per cell of round shape and an interlayer space with mucilage visible mainly in the lower and basal parts. In the basal and lower parts of most individuals, there are larger, colorless, modified cells with rhizoids for fixation to the substrate (Figure 6c,d). This species has been previously recorded from Africa, the Indian Ocean Islands, and the Middle East [27], therefore, this is the first record for Chile.

#### 2.2.3. *Ulva compressa* Linnaeus

It is characterized by light green tubular thallus, thin and soft to the touch, thickening towards the top. It adheres to the substrate through a small 0.3 ± 0.08 mm round disk. It has a monostromatic frond with oval cells, irregularly arranged along the entire length of the thallus (Figure 7).

On average, the total length of the thallus reaches 3.46 ± 0.88 cm with a basal width of 0.1 ± 0.03 cm, which reaches 0.26 ± 0.06 cm in the middle zone and 0.54 ± 0.14 cm at the top of the frond. In the cross-section, the thallus has an average thickness in the middle zone of 73.3 ± 2.9 µm, with cells 17.6 ± 3.6 µm high × 7 ± 1.1 µm in diameter (Figure 8). The chloroplasts are oval to round, occupying a large part of the cell. This species has a worldwide distribution, being reported in all oceans [27].

#### 2.2.4. *Ulva aragoensis* (Bliding) Maggs

It is characterized by a tubular, light green, and thin thallus that maintains its thickness from the disk to the apical zone and in some cases is slightly wider towards the apical zone (Figure 9). It is attached to the substrate by a small round disk that is 0.25 ± 0.097 mm in size and in some cases is irregularly shaped. It has monostromatic fronds with irregularly ordered cells in superficial view, while in the cross-section, the cells have a sub-square shape (Figure 10).

The average length of the thallus is 5.27 ± 1.32 cm. The basal width of the frond is 0.11 ± 0.030 cm; in the middle zone, the width is 0.16 ± 0.07 cm, and in the upper part, it is 0.16 ± 0.05 cm. In the cross-section, the average thickness of the frond in the middle zone is 26.14 ± 9.81 µm, with elongated cells 15 ± 2.2 µm high × 6.4 ± 0.8 µm in diameter (Figure 10). Oval and round chloroplasts occupy a large part of the cell. This species is widely distributed [27].

### 2.3. Molecular Phylogenetics

Partial sequences of the *ITS1* and *tuf*A genes were obtained for 42 and 13 individuals, respectively, of *Ulva* spp. from Algarrobo Bay, including DNA samples of individuals of five morphs: three with laminar, foliose, and distromatic fronds (morphs 2, 3, 4, and 4AM) and two with tubular, filamentous, and monostromatic fronds (morphs 100 and 200) (Table 1). The final *ITS1* and *tuf*A sequence alignments consisted of 370 and 732 bp including gaps, respectively. Phylogenetic analyses of *ITS1* showed five distinct groups of sequences with a variable degree of node support (Figure 11a). Each group of sequences formed was consistent with the morphological species identification. The topology is robust, with all but one of the groups formed with high Bayesian posterior probabilities (0.99–1) and bootstrap values greater than 80%. The phylogenetic tree of *tuf*A sequences from samples from Algarrobo revealed three groups (Figure 11b), only one of which has high posterior probability and bootstrap support. Two of the groups corresponded to morphological species identification while the other two morphs, 2 and 3, grouped together as one subdivided entity (in orange in Figure 11b), albeit without node support.

One sequence of each group of *Ulva* spp. from Algarrobo from the first round of phylogenetic analyses was used to blast against the GenBank database to determine the range of species to which they most closely matched. For further phylogenetic analyses, we included GenBank *ITS1* and *tuf*A sequences of *Ulva* that had been previously validated and used for the identification of *Ulva* species on different coasts worldwide (e.g., [32,34,35,36,37,46,49,50]). Sequences referred to as *Ulva* sp. A in [36] were considered as *U. uncialis* [48]. Species names and associated sequences that are synonyms were not included in the analysis even when blast searches closely matched them. Relevant cases involve the invalid names *U. pertusa* and *U. laetevirens*, which are synonyms of *U. australis* [51,52]; *U. fasciata*, a synonym of *U. lactuca*; and *U. stipitata*, a synonym of *U. fenestrata* [53]. Also, sequences incorrectly known as *U. rigida* that were from outside Europe and that correspond to *U. lacinulata* were not used [49]. With the above considerations, the final datasets consisted of 5 sequences from *Ulva* spp. from Algarrobo Bay (one from each group in Figure 11a) and 22 sequences of *Ulva* spp. from GenBank for *ITS1*, and 4 sequences from Algarrobo Bay (one from each group in Figure 11b) and 21 sequences from GenBank for *tuf*A (Table 2). There were no available *tufA* sequences of *U. stenophylloides*.

The *ITS1* phylogenetic reconstructions strongly suggest that the five groups of sequences detected from Algarrobo Bay samples correspond to different *Ulva* species (Figure 11c). Morph 2 grouped with *U. stenophylloides*, albeit with low branch support, phylogenetically close to morph 4, which grouped with *U. uncialis* and *U. lacinulata* with low bootstrap support but high Bayesian posterior probability (0.93). Morph 100 grouped with *U. compressa* with very high node support (99% bootstrap support and Bayesian posterior probability of 1). Morph 200 grouped closely with *U. aragoensis* with low branch support, within a group that also includes *U. flexuosa*, *U. torta*, and *U. californica*, supported by a 91% bootstrap value and a posterior probability of 1. Finally, morph 4AM grouped closely with *U. australis* with 96% bootstrap support and a posterior probability of 1.

Phylogenetic analyses of *tuf*A, albeit with lower taxonomic representation from Algarrobo Bay, confirm the presence of several species of *Ulva* in the samples with good support values for most relevant nodes (Figure 11d). Morphs 2 and 3 formed a group with relatively high node support. Morph 4 grouped with *U. uncialis* with 99% bootstrap support and a posterior probability of 1, with *U. lacinulata* as a sister taxon. Morph 4AM, as with the *ITS* marker, grouped with *U. australis* with high node support (100% bootstrap support and a posterior probability of 1).

Overall, both markers match morph 4 to *U. uncialis* and *U. lacinulata*, and morph 4AM to *U. australis*. We only had marker *ITS* for morphs 100 and 200, which grouped with *U. compressa* and *U. aragoensis*, respectively. Morph 2 is likely *U. stenophylloides* according to the *ITS* marker, and even though we could obtain sequences for *tuf*A, there are no sequences of *U. stenophylloides* in GenBank to compare with morph 2.

## 3. Discussion

The phylogenetic analyses of this study and our 2021–2022 field surveys across the seasons to quantify abundance and composition at Los Tubos beach in Algarrobo Bay, where green tides occur, showed that an *Ulva* species unreported until now in Chile (*U. stenophylloides*) dominated abundance throughout the year and tidal zones, followed by a succession of other *Ulva* species, which peak in abundance during different seasons. Moreover, our data suggest that *U. stenophylloides* is a generalist species regarding environmental conditions compared to other *Ulva* species and potentially a competitively dominant taxon over the other algae in the zone. *Ulva uncialis* showed considerable intertidal abundance (mean cover 42%) in winter, whereas it covered less than 10% in spring and summer, and it was almost absent in autumn, suggesting that the species may be more sensitive to environmental variation than *U. stenophylloides*. The other *Ulva* species (mostly with tubular thallus) were present with relatively high cover during spring, summer, and autumn but were almost absent in winter, suggesting that these may also be more sensitive to environmental conditions or that they have offset reproductive patterns. This type of offset seasonal changes among the species that form green tides have been observed worldwide (e.g., [47,56,57,58]) and deserves more detailed studies in Algarrobo. Some studies indicate that the degree of tolerance to a single factor (such as changes in temperature, light, the availability of a particular nutrient, anaerobic conditions, or other abiotic factors such as desiccation stress) would mostly explain these changes and the resistance and persistence of the green tides through seasons. In the case reported by Yoshida et al. [57], where *Ulva* species show different seasonal growth patterns in the green tide of Hiroshima Bay, it was concluded that temperature explained the differentiation in seasonal growth patterns. However, these authors also showed that other factors, such as seasonal variations in DIN (dissolved inorganic nitrogen), DIP (dissolved inorganic phosphorus), salinity, and irradiance, can also explain the seasonal growth patterns of *Ulva* species. As suggested by Wang et al. [5], higher levels of nitrate pollution caused by the rapid industrialization/urbanization in coastal zones would favor the rapid growth of *Ulva* species and promote the production of nitric oxide in *Ulva prolifera*, which further facilitates the differentiation of algal vegetative cells into reproductive cells and sporangia. This differentiation would raise the production of spores, further contributing to the formation of the green tide in the Yellow Sea. The thriving of local and invasive species of *Ulva* probably depends on the combined rise of nutrients, anaerobic conditions, and higher temperatures due to the anthropogenic intervention on the coast, which can vary significantly throughout the year. We suggest then that, in the context of anthropogenically impacted coastal ecosystems, the presence of *Ulva* species and relative abundances would likely depend on trade-offs in their ecophysiological potential, environmentally determined competitive abilities, or perhaps even range of tolerance to (or competitive capacities to exploit) some anthropogenic factors that may be more prevalent during some seasons than others. Future efforts should characterize the dominant oceanographic conditions (including the concentration of nutrients) and dominance of species along the coastal border of the bay through the seasonal cycle. A single study conducted in Algarrobo Bay on a spring day (October) 2013 corroborated the exposure of the bay to upwelling intensification but reported comparatively low nutrient concentrations (nitrates, phosphates) on surface waters (<5 m) during the sampling date and variable phytoplankton composition between the southern and northern tip of the bay [59]. Beyond this single one-day study, there are no records of synoptic or seasonal variability of nutrient concentration, circulation, or hydrographic conditions in Algarrobo Bay. Differences in reproductive patterns among the different *Ulva* species should also be investigated. On the other hand, as pointed out by Fort et al. [60], regarding the *Ulva* blooms from the coasts of Ireland and Brittany (France), the six *Ulva* species delineated by them through a comparison of their cytoplasmic genomes showed high levels of inter-specific genetic diversity and very rare occurrence of hybrids but still highly similar morphologies, which would agree with the existence of cryptic diversity but discrete and clear genetic separation among species in this genus. Nonetheless, in our study, the dominant species in Algarrobo Bay, *U. stenophylloides* and *U. uncialis*, showed clear morphological distinctiveness and a contrasting difference in the toughness of their blades when manipulated in the field, with *U. uncialis* having thicker and structurally stronger blades than *U. stenophylloides*, which displays a considerably softer tissue and less stretch resistance of its body structure. *Ulva* species can experience species turnover across seasons or years due to spatiotemporal environmental heterogeneity or trade-offs between fecundity, survival, and dispersal [61]. In addition, the opposing forces of ecological drift and negative frequency dependence (rare-species advantage) could jointly shape coexistence [62] in this *Ulva* assemblage.

The spatiotemporal monitoring and the morphological and sequence-based analysis carried out in Algarrobo Bay allowed the identification of at least five species of the genus *Ulva*. Consequently, we can conclude that the green tides present on the coast of Algarrobo in Chile are multispecific. Although one species is dominant, *U. stenophylloides*, depending on the season, the green tides exhibit different relative abundances. For instance, during winter, *U. stenophylloides* and *U. uncialis* reach similar abundances. In spring and autumn, *U. stenophylloides* prevails, with a lesser presence of *U. uncialis*, *U. compressa*, and *U. aragoensis*. Although visual positive identification is difficult, the monitoring over different seasons, together with the molecular data, has allowed to us recognize and differentiate the specimens present in the field based on their morphological and textural attributes, such as the habit of the frond, the thickness of the thallus, and the presence of a short stipe. This is very significant because at least a preliminary assessment of the types of *Ulva* present in incipient green tides on different shores along the country could be made. For instance, *U. uncialis* has a lamellar thallus, thicker and stiffer to the touch, attached to the substrate via a short stipe. In contrast, the other species of lamellar thallus *U. stenophylloides* has a softer and thinner thallus, with almost transparent fronds and an inconspicuous stipe. The other two cylindrical-thallus species, *U. compressa* and *U. aragoensis*, could also be distinguished via thallus morphology. *Ulva compressa* has a thicker thallus, and the fronds tend to be compressed or flattened along the thallus, whereas *U. aragoensis* has a thinner thallus, and its fronds are always cylindrical along their entire length. These morphological characteristics agree with previous work describing these taxa either in Chile or worldwide (e.g., [24,25,26,32,39,42,63]). Nonetheless, even though some tubular species were clearly identified molecularly [64] and morphologically [65] as is the case with *U. compressa* and *U. aragoensis* in our study, difficulty still persists in identifying some other tubular *Ulva* species such as those classified under the LPP (linza–procera–prolifera) complex [65,66]. This highlights the fact other tubular non-monostromatic and less abundant *Ulva* species probably remain to be discovered or clearly identified at Algarrobo and worldwide, due to the need to genetically characterize all available specimens of distromatic foliose taxa [67] and the need to design more universal DNA primers that allow the reproducible sequencing of most of the existing *Ulva* species.

Despite all the efforts made to achieve certainty regarding the taxonomic status of *Ulva* species, confusion persists and puts in doubt their taxonomic nomenclature. Thus, the taxonomy and nomenclatural history of each taxon becomes increasingly complicated, especially for species with names that have been repeatedly applied worldwide, such as *Ulva lactuca* L. and *Ulva rigida* C. Agardh (e.g., [67]). In the case of *U. rigida*, Hughey et al. [49] sequenced and analyzed the lectotype of this species from Cádiz, Spain, finding that it would be restricted in origin and distribution to the coasts of Ireland, Spain, and Portugal, and would not be widely distributed. The authors postulate that the material identified as “*U. rigida*” in GenBank from other seas, including those of Chile, is erroneously classified and would instead belong to *U. lacinulata* (Kutzing) Wittrock. On the other hand, *Ulva stenophylloides* has been previously described and reported for Victoria, Australia [32]; New Zealand [68]; and South Africa [48]. Hence, our contribution would be the first record of *U. stenophylloides* in Chile, as well as in the case of *U. uncialis* and *U. aragoensis*. However, more ecological and physiological studies, including fitness studies, are required to corroborate the presence of these five species along the Chilean coast.

DNA sequences of two specific genomic regions, *tuf*A and *ITS*, have been widely used for species identification across taxa (see introduction). However, species identification based solely on sequences can be ambiguous because of the sequence similarity and high intraspecific variation. Additionally, species names associated with data on available databases can be misannotated (e.g., [67,69]), and care should be taken when incorporating database searches into analyses, and researchers should consider including the detected variability in the results. Although molecular markers alone do not always allow precise species identification, in many cases, they provide useful information for the identification of green tide species (e.g., [47]). Herein, we took an interdisciplinary approach to determine which taxa were present in the study site, including ecological aspects, morphological examination, and molecular data analyses. This is an important lesson for *Ulva* and other species delineation, to undertake an interdisciplinary approach including ecological data, morphological analyses, and DNA sequences, with the sequence data complementing the morphological and ecological identification. Taken together, *ITS1* and *tuf*A phylogenetic analyses are congruent with the morphological assessment of the morphs of *Ulva* present in the green tides of Algarrobo in this study; the use of methods based on morphology and molecular markers proved to be an approach that allowed recognition of taxonomic diversity. Further studies, including more sampling sites and individuals per site, could allow finer analysis, such as assessments of the connectivity, degree of local genetic diversity, and phylogeographic structure of the *Ulva* species detected in the green tides of the Algarrobo area. Also, physiological studies and multivariate approaches are needed to understand the proliferation of these species along Algarrobo Bay and other parts of the Chilean coast.

## 4. Conclusions

We provide the first report on the morphological and molecular biodiversity of the *Ulva* spp. making up the green tides occurring in Chile, on the central coast (Algarrobo Bay). Our results indicate that green tides in this area are a multi-species phenomenon involving at least five different taxa, with variations in coverage among seasons and sections of the intertidal zone. This study provides a basis for the development of managing efforts at the socioecological, ecophysiological, population, and phylogeographic levels, as well as for seaweed aquaculture and biotechnology applications in the country.

## 5. Materials and Methods

### 5.1. Sampling

Sampling of *Ulva* spp. was performed during low tides along 200 m of the rocky intertidal zone of Los Tubos beach in Algarrobo Bay, located south of Valparaíso, Chile (33°21.891′ S, 71° 40.763′ W). This site is characterized by a sedimentary rocky platform that slopes gently towards the sea, with channels and heterogeneity produced mainly by granitic rock boulders. First, preliminary samplings were conducted during the summer of 2020 and 2021 to identify the morphological diversity of *Ulva* spp. along the Algarrobo Bay. Posteriorly, robust sampling conducted in the winter and spring of 2021 and during the summer and autumn of 2022 was realized for the morphological and molecular characterization of the specimens and their cover along the intertidal zone. Particularly, we used three transects, which spanned throughout the entire intertidal section. Along the transects, we recorded the presence of the different *Ulva* morphotypes, going from the upper to the lower intertidal zone. Plants representing the different morphologies were carefully removed from the substratum using a spatula and placed in individually labeled bags. Additionally, cover was recorded using the point intercept method with 10–60 reticulated 0.25 m^2^ quadrats per transect and along the high, middle, and low intertidal zones. Given the results, in this study, we separated the *Ulva* spp. complex into three groups: *U. stenophylloides*, *U. uncialis*, and *Ulva* spp.

### 5.2. External and Internal Morphology

The collected algae were differentiated first through the analysis of external morphology, based on attributes such as the size of the blades and the color, shape, and texture of the thallus. Characteristics of the blade’s edge and the presence or absence of perforations were also considered. Regarding the algae size, the width at different heights of the thallus (basal, middle, and upper parts) and the size of the longest blade were measured. As for the internal morphological attributes, the shape and arrangement of the cells were recorded by superficial observation of the tissues under a microscope, at different heights of the thallus (basal, middle, and upper parts). Additionally, through a cross-section in different parts of the thallus as previously mentioned, the width and height of the cells, the thickness of the thallus, and the number of cell layers were measured using an inverted microscope (Eclipse Ts2, Nikon, Japan) and ImageJ software (https://imagej.net/ij/download.html, accessed on 26 April 2024). The shape of the cells, the presence or absence of interlayer content, the shape and chloroplast disposition, the number of pyrenoids, and the origin and disposition of rhizoids were also considered and noted for comparison.

### 5.3. Sampling for Molecular Analyses

A total of ca. 100 *Ulva* specimens collected along the intertidal zone were stored immediately in plastic bags containing seawater and transported to the laboratory (www.lebma.cl, accessed on 26 April 2024) in a cooler at 5–7 °C. Later, algae were rinsed with 0.22 μm filtered seawater and immediately pressed as herbarium vouchers. Subsamples were dried in desiccant silica gel (Vetec Analytical Reagents, Brazil) for subsequent DNA analysis, and other subsamples were kept at 4 ± 1 °C prior to morphological analysis. The specimens (17) are housed in the herbarium of the National Museum of Natural History, Chile, under the numbers SGO 171635–171651.

### 5.4. DNA Extraction and PCR Amplification

The fronds were finely ground in liquid nitrogen, and DNA was extracted following the protocol described by Saunders [70], with modifications [71].

The nuclear marker *ITS1* of 45S rDNA repeats was amplified using the *ITS1* primers ITS1F (5′TCGTTGAACCCTCCCGTTTA’3) and ITS1R (5′CGATGACTCACGGAATTCTGC’3). The plastid elongation factor *tuf*A was amplified using the primers tufGF4 (5′ GGNGCNGCNCAAATGGAYGG 3′) and tufAR (5′ CCTTCNCGAATMGCRAAWCGC 3′) [35]. Additionally, our newly designed primers tufAMF2 (5′ CGGGTKCTGATGGTCCTATGC 3′) and tufAMR1 (5′ CTCGATCTCCTGGRATTACC 3′) were created using as reference sequences of *Ulva rigida* (accession No. MT078951.1) and *Ulva prolifera* (accession No. EF595302.1), and our reverse primer tufAMR1 (5′ CTCGATCTCCTGGRATTACC 3′) was newly created using as reference sequences of *Ulva australis* (accession No. MF172079.1) and *Ulva rigida* (accession No. MT078951.1).

The PCRs were performed in a 2720 Thermal Cycler Applied Biosystems™ (AB, Singapore), with a final volume of 15.5 µL per reaction: 1 μL of diluted DNA (60 ng/µL), 3.0 μL of Buffer Go Taq^®^, 0.075 μL of 0.1% BSA (1 mg mL^–1^), 0.3 μL of dNTP mix (2.5 mM), 0.12 μL of MgCl_2_ (2.5 mM), 1.5 μL of forward and reverse primers (10 µM), 0.15 μL of Go Taq^®^, and 9.35 μL of ddH_2_O. PCR profiles were as follows: *ITS1*: an initial 3 min denaturation at 95 °C, 30 cycles of 95 °C for 30 s, 51 °C annealing for 1.5 min, and 72 °C extension for 1.5 min, followed by 72 °C final extension for 7 min; *tuf*A: an initial 5 min denaturation at 95 °C, 30 cycles of 95 °C for 30 s, 43 °C annealing for 30 s, and 72 °C extension for 40 s, followed by 72 °C final extension for 7 min. It is worth noting that for a few difficult samples, low melting temperatures (from 30 °C to 37 °C) were tested for both markers and produced good results. All PCR products were purified using UltraCleanTM DNA Purification kits (MO BIO Laboratories, Carlsbad, CA, USA) and sequenced using the forward and reverse amplification primers by Pontificia Universidad Católica de Chile (https://sites.google.com/bio.puc.cl/omics/secuenciacion-puc-chile, accessed on 1 March 2022).

Sequences obtained using forward and reverse primers of each individual and marker (*ITS1* and *tuf*A) were aligned to obtain a simple sequence using the Geneious R10 (www.geneious.com, accessed on 26 April 2024) MAFFT alignment plugin [72] using the Q-INS-I iterative method. Alignments were visually inspected in Geneious R10 and translated to amino acids to verify the absence of stop codons. Sequences were deposited in GenBank under accession numbers OR514517–58 and OR526355–67 for *ITS1* and *tuf*A, respectively (Appendix A). A second alignment was similarly built for each marker with sequences of *Ulva* spp. from Algarrobo along with GenBank [73] sequences of *Ulva* spp. that were highly similar in BLAST searches and that had been previously used to identify *Ulva* species [74,75]. Modeltest [76], as implemented in PAUP’s plugin [77] for Geneious R10, was used to select the model of molecular evolution for each dataset. Phylogenetic trees were inferred using Maximum Likelihood, with mid-point rooting as well as branch and bound tree searching in PAUP and using Bayesian approximation with the MrBayes [78,79] plugin for Geneious R10. Support for nodes was estimated using 1000 bootstrap replicates with heuristic search with Maximum Likelihood along with posterior probabilities estimated using 10,000,000 iterations and 20% of the trees discarded as burn-in from Bayesian analyses.

## Figures and Tables

**Figure 1 plants-13-01258-f001:**
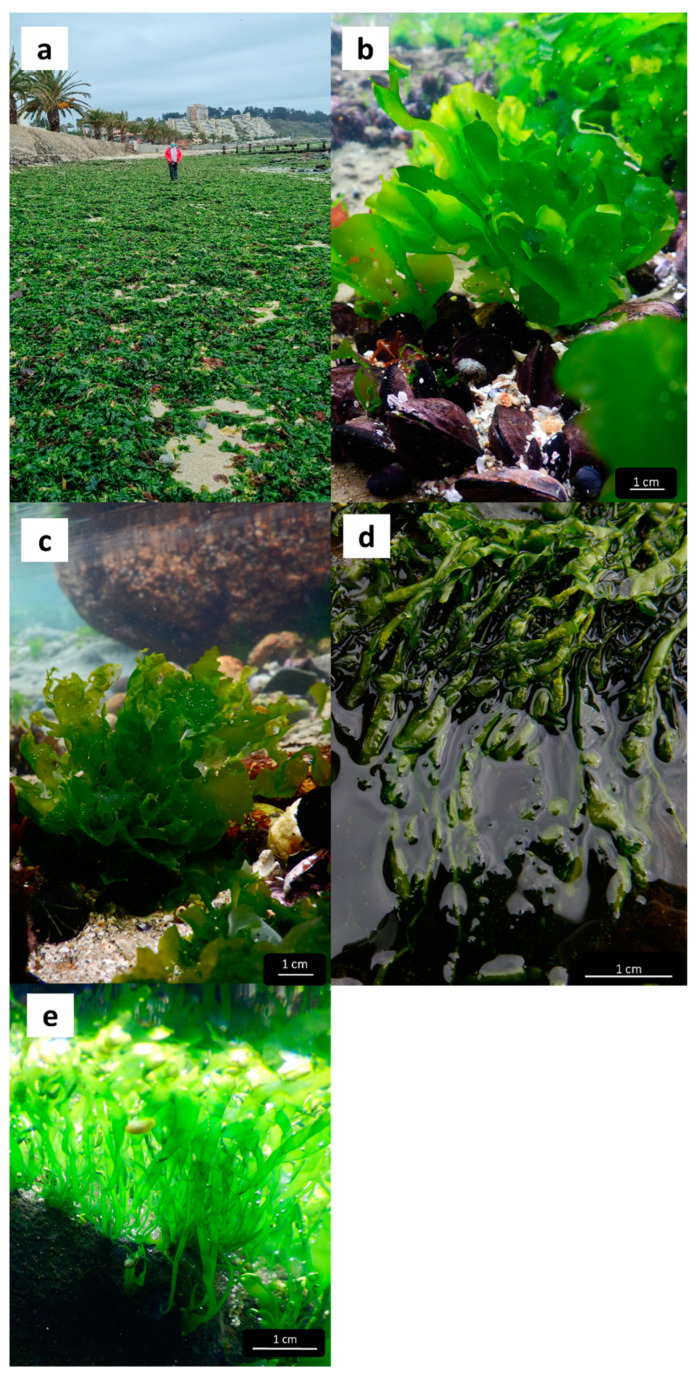
Green tide blooms (**a**) and habit of the *Ulva* spp. from Los Tubos beach, Algarrobo Bay, Chile: (**b**) *U. stenophylloides*, (**c**) *U. uncialis*, (**d**) *U. compressa*, (**e**) *U. aragoensis*.

**Figure 2 plants-13-01258-f002:**
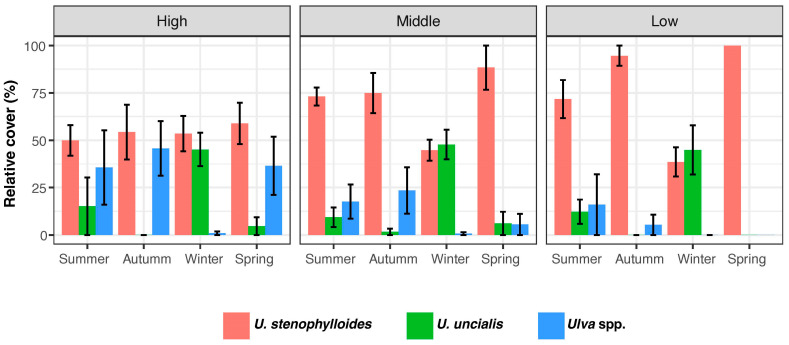
*Ulva* species relative cover on Los Tubos beach, Algarrobo, along the intertidal zones (high, middle, low) for each season in this study. Bars correspond to the mean relative cover, and the bars show the standard error.

**Figure 3 plants-13-01258-f003:**
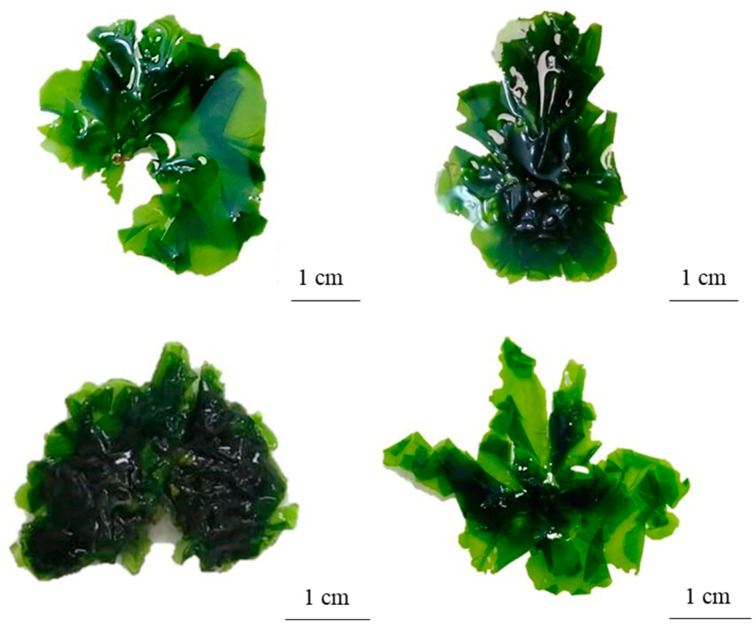
Morphological variation of the thallus of *Ulva stenophylloides*.

**Figure 4 plants-13-01258-f004:**
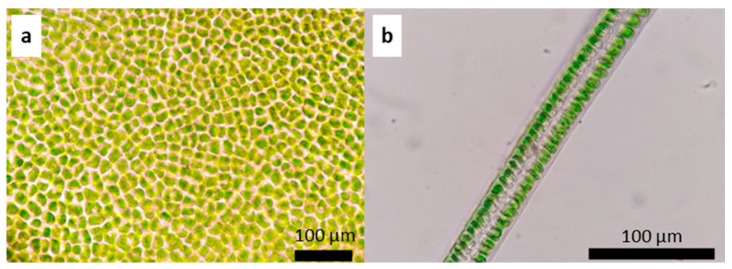
*Ulva stenophylloides*: (**a**) surface image and (**b**) cross-section of the middle zone of the thallus.

**Figure 5 plants-13-01258-f005:**
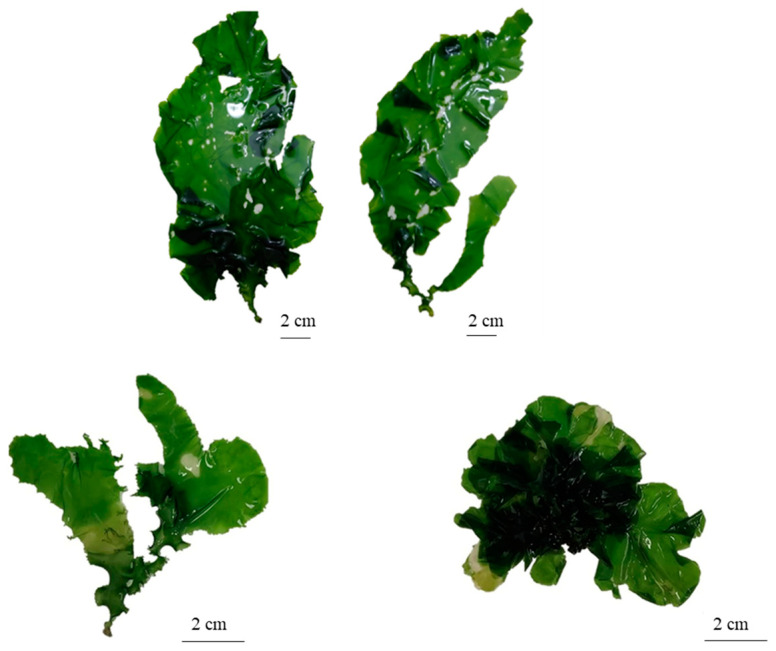
Morphological variation of the thallus of *Ulva uncialis*.

**Figure 6 plants-13-01258-f006:**
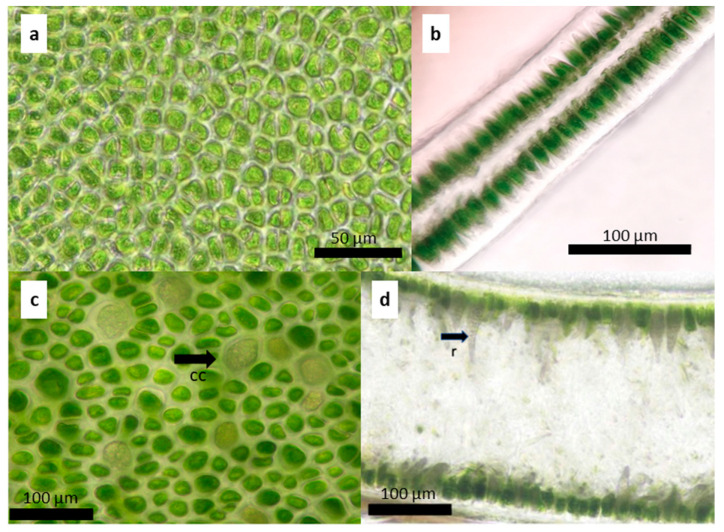
*Ulva uncialis*: (**a**) surface image and (**b**) cross-section of the middle zone of the thallus. The lower zone of the thallus, presence of rhizoidal basal cells: (**c**) superficial image of colorless cells in the superficial view (cc) and (**d**) cross-section where the rhizoids are observed (r).

**Figure 7 plants-13-01258-f007:**
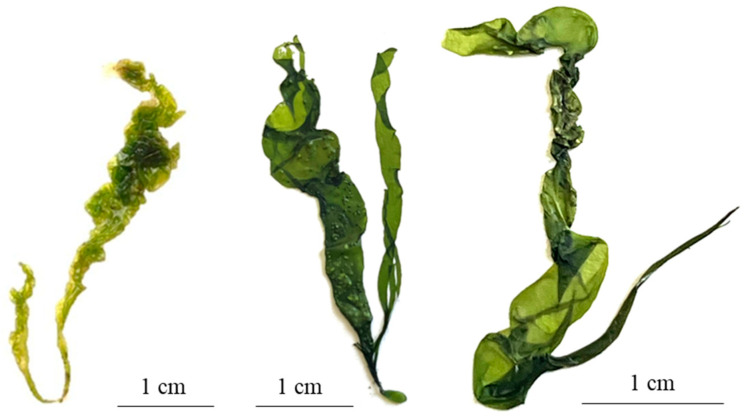
Thallus of *Ulva compressa*.

**Figure 8 plants-13-01258-f008:**
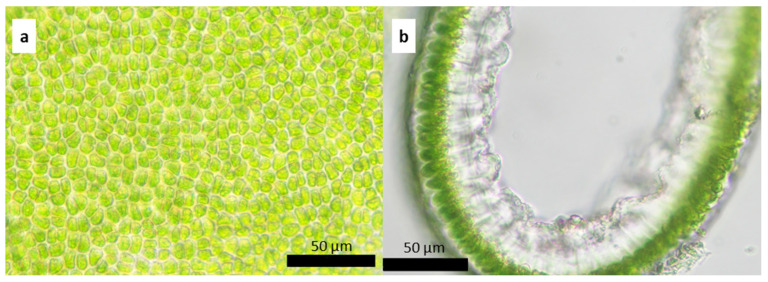
*Ulva compressa*: (**a**) surface image and (**b**) cross-section of the middle zone of the thallus.

**Figure 9 plants-13-01258-f009:**
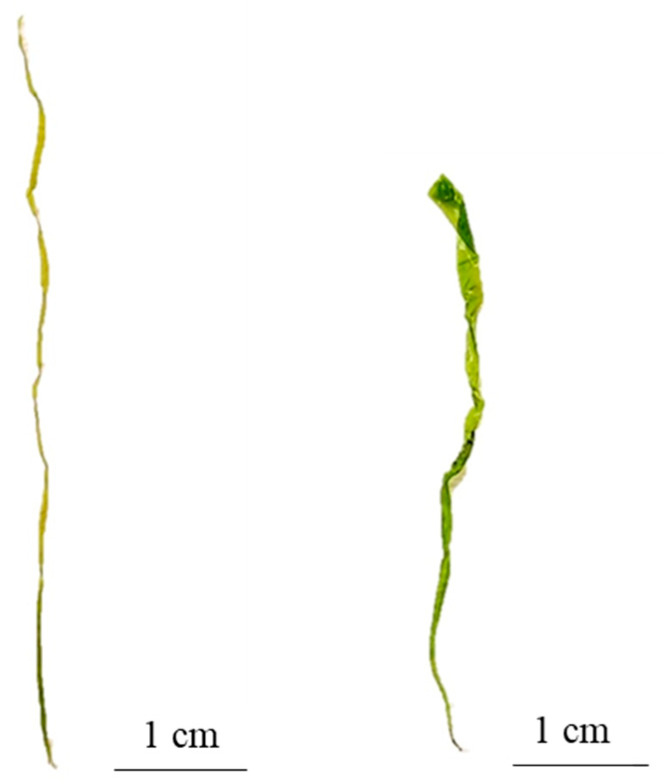
Thallus of *Ulva aragoensis*.

**Figure 10 plants-13-01258-f010:**
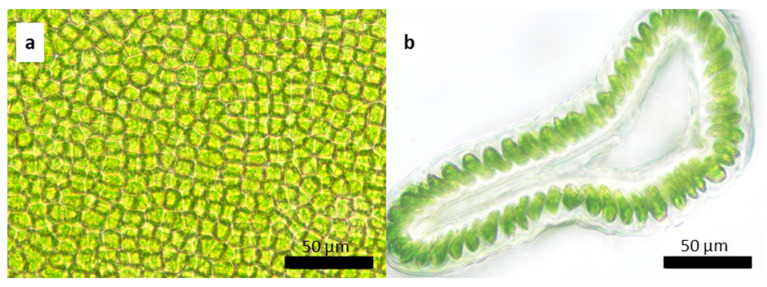
*Ulva aragoensis*: (**a**) surface image and (**b**) cross-section of the middle zone of the thallus.

**Figure 11 plants-13-01258-f011:**
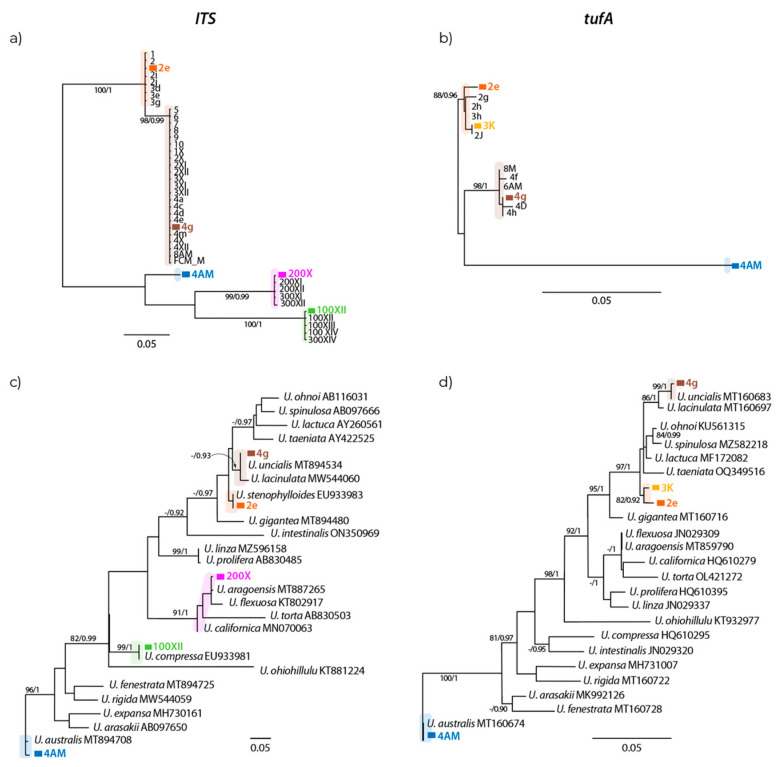
Maximum Likelihood phylograms of *ITS1* and *tuf*A sequences of *Ulva* from Algarrobo Bay ((**a**,**b**), respectively) and for the same markers including sequences from GenBank of other *Ulva* species (**c**,**d**). Support values of nodes correspond to Maximum Likelihood bootstrap values greater than 80% and Bayesian posterior probabilities greater than 0.9; “-” indicates lower values. Branches without support values have bootstrap values lower than 80% and posterior probabilities lower than 0.9. Colors designate different morphs.

**Table 1 plants-13-01258-t001:** External characteristics of the different *Ulva* species recognized on Los Tubos beach, Algarrobo Bay, Chile.

Character	*Ulva stenophylloides* (*morph 2*)	*Ulva uncialis* (*morph 4*)	*Ulva compressa* (*morph 100*)	*Ulva aragoensis* (*morph 200*)
Color	Intense to light green	Dark green	Light green	Light green
Thallus shape	Laminar foliosus	Laminar foliosus	Cylindrical	Cylindrical
Texture	Smooth and thin	Smooth and thick	Smooth and thin	Smooth and thin
Algae edge (upper)	Smooth and undulated	Smooth	Smooth	Smooth
Seaweed edge (middle)	Smooth and undulated	Wavy	Smooth	Smooth
Seaweed edge (basal)	Smooth and serrated	Toothed and with excrescences	Smooth	Smooth
Disk shape	Round	Round	Round	Round to irregular
Perforations in the fronds	Present	Present	Absent	Absent

**Table 2 plants-13-01258-t002:** Species and GenBank accession numbers used for phylogenetic analyses with *ITS1* and *tuf*A sequences of *Ulva* from Algarrobo Bay.

Species	*ITS1*		*tuf*A	
	GenBank Accession Number	References	GenBank Accession Number	References
*U. aragoensis*	MT887265	[35,42]	MT859790	[42,46]
*U. arasakii*	AB097650	[36]	MK992126	[36]
*U. australis*	MT894708	[36]	MT160674	[36,46]
*U. californica*	MN070063	[34,35]	HQ610279	[37,46]
*U. compressa*	EU933981	[32]	HQ610295	[37,46]
*U. expansa*	MH730161	[36]	MH731007	[36,52,53]
*U. fenestrata*	MT894725	[36]	MT160728	[36,46]
*U. flexuosa*	KT802917	[35]	JN029309	[43,46]
*U. gigantea*	MT894480	[36]	MT160716	[36]
*U. intestinalis*	EU933936	[32]	JN029320	[43,46]
*U. lacinulata*	MW544060	[36]	MT160697	[36,46]
*U. lactuca*	AY260561	[36]	MF172082	[36]
*U. linza*	MZ596158	[35]	JN029337	[43,46]
*U. ohiohillulu*	KT881224	[36]	KT932977	[36]
*U. ohnoi*	AB116031	[36]	KU561315	[54]
*U. prolifera*	AB830485	[35]	HQ610395	[37,46]
*U. rigida*	MW544059	[36,49]	MT160722	[36]
*U. spinulosa*	AB097666	[34]	MZ582218	[44]
*U. stenophylloides*	EU933983	[32]	*	
*U. taeniata*	AY422525	[34]	OQ349516	[50]
*U. torta*	AB830503	[34]	OL421272	[46,55]
*U. uncialis*	MT894534	[36]	MT160683	[36]

* not available.

## Data Availability

Data are contained within the article.

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
