# Peer review of "Morphological and Molecular Identification of Ulva spp. (Ulvophyceae; Chlorophyta) from Algarrobo Bay, Chile: Understanding the Composition of Green Tides"

_plants, 2024, doi:10.3390/plants13091258_

Round 1

Reviewer 1 Report

Comments and Suggestions for Authors

The Authors provide a significant contribution to the understanding of the blooms of Chilean Ulva species, based on a combination of ecological, morpho-anatomical, and molecular information. They recognize, both on a morphological and molecular basis, 5 distinct taxa. Of them, three (U. stenophylloides, U. rigida, U. australis) have laminar, foliose, and distromatic morphology, while the other two (U. compressa, U. flexuosa) have tubular, filamentous, and monostromatic fronds.

The morphological description of the Chilean specimens, the iconography produced, as well as the ecological data and the socio-economic repercussions of the bloom events are well done.

Nevertheless, since that significant progress has been made in the last decade to apply the correct names to Ulva species by sequencing type specimens or type material (see: Hanyuda & Kawai 2018; Hughey et al. 2018, 2019, 2021, 2022; Hughey & Gabrielson 2022; Hughey et a.l 2024), the authors should consider in their analysis of molecular data only the sequences on such type material published  in the aforementioned publications, in order to ensure a correct identification of their specimens. Comparing sequences deposited in GenBank of specimens of doubtful identification, creates further confusion.

Consequently, if after re-elaboration of the molecular results, which I strongly suggest, the identification is confirmed, greater emphasis in the discussion should be given to the species present in Chile, in particular Ulva rigida, previously excluded from Chile and U. stenophylloides (or Ulva taeniata?)  given that it would represent the first documented reports in this area.

Therefore, in my opinion the manuscript can be accepted only after the re-elaboration of the molecular data in order to ascertain the correct identification of the Chilean specimens in light of the most recent bibliography regarding this genus.

Author Response

Reviewer 1

The Authors provide a significant contribution to the understanding of the blooms of Chilean Ulva species, based on a combination of ecological, morpho-anatomical, and molecular information. They recognize, both on a morphological and molecular basis, 5 distinct taxa. Of them, three (U. stenophylloidesU. rigidaU. australis) have laminar, foliose, and distromatic morphology, while the other two (U. compressaU. flexuosa) have tubular, filamentous, and monostromatic fronds.

The morphological description of the Chilean specimens, the iconography produced, as well as the ecological data and the socio-economic repercussions of the bloom events are well done.

Nevertheless, since that significant progress has been made in the last decade to apply the correct names to Ulva species by sequencing type specimens or type material (see: Hanyuda & Kawai 2018; Hughey et al. 2018, 2019, 2021, 2022; Hughey & Gabrielson 2022; Hughey et a.l 2024), the authors should consider in their analysis of molecular data only the sequences on such type material published  in the aforementioned publications, in order to ensure a correct identification of their specimens. Comparing sequences deposited in GenBank of specimens of doubtful identification, creates further confusion.

Consequently, if after re-elaboration of the molecular results, which I strongly suggest, the identification is confirmed, greater emphasis in the discussion should be given to the species present in Chile, in particular Ulva rigida, previously excluded from Chile and U. stenophylloides (or Ulva taeniata?)  given that it would represent the first documented reports in this area.

Therefore, in my opinion the manuscript can be accepted only after the re-elaboration of the molecular data in order to ascertain the correct identification of the Chilean specimens in light of the most recent bibliography regarding this genus.

Thank you very much, we sincerely appreciate the time you spent reviewing this manuscript and the comments you made to improve it. We use track changes to highlight the modifications along the text.

As requested by reviewer we re-run phylogenetic analyses only considering previous advances in terms of unambiguous species names associated to used sequences from GenBank and avoid doubtful identification of the sequenced Algarrobo Bay specimens.

Specifically, for ITS we used all sequences included in Fort et al. (2022) and sequences published by Kraft et al (2010), as well as sequences used in phylogenetic analyses of Kang et al. 2019 and Xia et al. 2023, among others. For phylogenetic analyses based on tufA we also used sequences in Fort et al 2022, excepting reported accession number for U. ohnoi because it was wrongly assigned as a tufa sequence (we used a sequence of tufa for this species published by Melton III et al 2016. We also included sequences validated by Streinhagen et al. 2023, Tran et al 2023, and Hughey et al 2024, among others. The methods have been changed accordingly and new Table 2 was included with GenBank accessions and references associated to each of the sequences used in the phylogenetic analyses.

The sequence of the morph 4 (4g) was identified as U. uncialis (Ulva sp. A in Fort et al. 2022 identified as U. uncialis by Bachoo et al. 2023). And U. flexuosa as U. aragoensis (Krupnik e al. 2018). The other identifications for the other Ulva species agreed with our previous phylogenetic analyses.

Reviewer 2 Report

Comments and Suggestions for Authors

This paper is the first report on species composition of green tide in Chilean coastal area with systematic analysis using morphological and molecular information. Also, elucidation of species composition is the first step to develop countermeasures to green tides which causes many ecological and social problems in coastal areas worldwide. I recommend this paper to be published in the journal Plants. Manuscript is well written in good order and understandable. However, I want some points to be reconsidered by the authors. Consider the following and make corrections or add information before publishing.

 L74-75

 “Ulva species have been highlighted for rapidly increasing their abundance under certain forms of pollution (e.g., heavy metals)・・”

Are there any papers that show evidence that heavy metal pollution promotes the growth of Ulva and causes green tides? Eutrophication can cause green tide occurrence, but heavy metal pollution can also cause green tides?

L133 Do the various forms shown in Table 1 match the type specimens and the forms of each species studied in other countries? As described below, sequence information alone is not necessarily sufficient for species identification. Morphological comparisons and examinations with type specimens possibly could make species identification more certain.

L140 2.1 Ulva spp covers

     It is better to indicate the relative height of high, middle, and low intertidal zone (for example, height from the Lowest Low-water Level).

The survey of % cover seems to be based on visual observation, but it seems that U.stenophylloides and U. rigita are very similar in morphology and are thought to be extremely difficult to distinguish in the field. How were the two species distinguished in the field surveys?

L236 2.3 Molecular phylogenetics

 Table 2

The species of the sample is determined based on the high degree of similarity in the DNA sequence with those in the database. Sample 4g (Morph 4 = Ulva rigida) also has high similarity to Ulva uncialis, but what was the reason for determining that it was Ulva rigida rather than Ulva uncialis?

 Regarding turfA, there is no information on U. stenophyylloides, and turfA could not be amplified by PCR with samples 100X and 200X. Therefore, the judgment is based only on the results of ITS1 regions, but is that sufficient? Isn't it necessary to add information from other regions, such as COI, when making decisions?

Comments on the Quality of English Language

English is well-written and very understandable.

Author Response

Reviewer 2

This paper is the first report on species composition of green tide in Chilean coastal area with systematic analysis using morphological and molecular information. Also, elucidation of species composition is the first step to develop countermeasures to green tides which causes many ecological and social problems in coastal areas worldwide. I recommend this paper to be published in the journal Plants. Manuscript is well written in good order and understandable. However, I want some points to be reconsidered by the authors. Consider the following and make corrections or add information before publishing.

Thank you very much, we sincerely appreciate the time you spent reviewing this manuscript and the comments you made to improve it. We use track changes to highlight the modifications along the text.

L74-75

“Ulva species have been highlighted for rapidly increasing their abundance under certain forms of pollution (e.g., heavy metals)”

Are there any papers that show evidence that heavy metal pollution promotes the growth of Ulva and causes green tides? Eutrophication can cause green tide occurrence, but heavy metal pollution can also cause green tides?

 We included a reference.  Ge, C.; Yu, X.; Kan, M.; Qu, C. Adaption of Ulva pertusa to multiple-contamination of heavy metals and nutrients: Biological mechanism of outbreak of Ulva sp. green tide. Mar Poll. Bull. 2017, 125, 250–253

L133 Do the various forms shown in Table 1 match the type specimens and the forms of each species studied in other countries? As described below, sequence information alone is not necessarily sufficient for species identification. Morphological comparisons and examinations with type specimens possibly could make species identification more certain.

 Yes, we modified certain statements for better compression.

L140 2.1 Ulva spp covers

     It is better to indicate the relative height of high, middle, and low intertidal zone (for example, height from the Lowest Low-water Level).

We separate the intertidal according to the distribution of the species (e.g. seaweeds and/or animals), the zone is a platform with low o null inclination.

The survey of % cover seems to be based on visual observation, but it seems that U.stenophylloides and U. rigita are very similar in morphology and are thought to be extremely difficult to distinguish in the field. How were the two species distinguished in the field surveys?

 During the field work all the samples were observed using magnifying glasses and posteriorly microscopy. The field work was realized by 7 persons, all with experience in algal taxonomy (Dr. Nicolas Latorre, Msc Alejandra Núñez, Dr (c) Florentina Piña, Mar Biol. Javiera Mutizabal, Dra. Loretto Contreras-Porcia, Msc María Eliana Ramírez, Dr. Andrés Meynard). The major part of the human equip are professors in botany and phycology. Before this ms, we worked in Algarrobo recollecting Ulva spp during the blooms during 2021 and 2022, where the specimens were analysed in our laboratory. So, we work adequately for the coverage determination. The morpho are very different such as was described along the ms. But we understand the point of the reviewer.

L236 2.3 Molecular phylogenetics

 Table 2

The species of the sample is determined based on the high degree of similarity in the DNA sequence with those in the database. Sample 4g (Morph 4 = Ulva rigida) also has high similarity to Ulva uncialis, but what was the reason for determining that it was Ulva rigida rather than Ulva uncialis?

 Regarding turfA, there is no information on U. stenophyylloides, and turfA could not be amplified by PCR with samples 100X and 200X. Therefore, the judgment is based only on the results of ITS1 regions, but is that sufficient? Isn't it necessary to add information from other regions, such as COI, when making decisions?

We re-run phylogenetic analyses only considering previous advances in terms of unambiguous species names associated to used sequences from GenBank and avoid doubtful identification of the sequenced Algarrobo Bay specimens.

Specifically, for ITS we used all sequences included in Fort et al. (2022) and sequences published by Kraft et al (2010), as well as sequences used in phylogenetic analyses of Kang et al. 2019 and Xia et al. 2023, among others. For phylogenetic analyses based on tufA we also used sequences in Fort et al 2022, excepting reported accession number for U. ohnoi because it was wrongly assigned as a tufa sequence (we used a sequence of tufa for this species published by Melton III et al 2016. We also included sequences validated by Streinhagen et al. 2023, Tran et al 2023, and Hughey et al 2024, among others. The methods have been changed accordingly and new Table 2 was included with GenBank accessions and references associated to each of the sequences used in the phylogenetic analyses.

The sequence of the morph 4 (4g) was identified as U. uncialis (Ulva sp. A in Fort et al. 2022 identified as U. uncialis by Bachoo et al. 2023). And U. flexuosa as U. aragoensis (Krupnik e al. 2018). The other identifications for the other Ulva species agreed with our previous phylogenetic analyses.

For non-foliose Ulva species there is less consensus as to which sequences should be used for species identification and phylogenetic analyses. Instead for foliose species, Fort et al. (2022) have identified submission numbers to correctly identify species.

We were hoping to amplify three DNA regions for each Ulva morph detected. Unfortunately, we were not successful with rcbl. For tufa we had only a partial success, limiting the power of the phylogenetic analysis for those morphs for which we obtained only one sequence marker.  Unfortunately, there are no COI sequences of U. stenophylloides in GenBank to be used as an alternative marker for species identification. However, we did not only rely on DNA sequences for identification, but we also performed exhaustive morphological analyses to independently identify the morphs based on morphology.

Round 2

Reviewer 1 Report

Comments and Suggestions for Authors

he molecular data have been reviewed by the authors in accordance with the latest bibliography on this topic, considering the suggestions. The specific attributions of the Chilean specimens have been changed because of this. The text has been revised in the light of the new results. 

Author Response

Thank you for the review, your input was very valuable.